# Identification of people with low prevalence diseases in administrative healthcare records: A case study of HIV in British Columbia, Canada

Scott D. Emerson[1]*, Taylor McLinden[1,2], Paul Sereda[1], Viviane D. Lima[1,3], Robert S. Hogg[1,2], Katherine W. Kooij[1,2], Amanda M. Yonkman[1], Kate A. Salters[1,2], David Moore[1,3], Junine Toy[1], Jason Wong[4], Theodora Consolacion[4], Julio S. G. Montaner[1,3], Rolando Barrios[1,5]

1 British Columbia Centre for Excellence in HIV/AIDS, Vancouver, British Columbia, Canada, 2 Faculty of Health Sciences, Simon Fraser University, Burnaby, British Columbia, Canada, 3 Department of Medicine, Faculty of Medicine, University of British Columbia, Vancouver, British Columbia, Canada, 4 British Columbia Centre for Disease Control, Vancouver, British Columbia, Canada, 5 School of Population and Public Health, University of British Columbia, Vancouver, British Columbia, Canada

* semerson@bccfe.ca

**Data Availability Statement:** The BC-CfE is prohibited from making individual level data

## Abstract

### Introduction

Case-finding algorithms can be applied to administrative healthcare records to identify people with diseases, including people with HIV (PWH). When supplementing an existing registry of a low prevalence disease, near-perfect specificity helps minimize impacts of adding in algorithm-identified false positive cases. We evaluated the performance of algorithms applied to healthcare records to supplement an HIV registry in British Columbia (BC), Canada.

### Methods

We applied algorithms based on HIV-related diagnostic codes to healthcare practitioner and hospitalization records. We evaluated 28 algorithms in a validation sub-sample of 7,124 persons with positive HIV tests (2,817 with a prior negative test) from the STOP HIV/AIDS data linkage–a linkage of healthcare, clinical, and HIV test records for PWH in BC, resembling a disease registry (1996–2020). Algorithms were primarily assessed based on their specificity–derived from this validation sub-sample–and their impact on the estimate of the total number of PWH in BC as of 2020.

### Results

In the validation sub-sample, median age at positive HIV test was 37 years (Q1: 30, Q3: 46), 80.1% were men, and 48.9% resided in the Vancouver Coastal Health Authority. For all algorithms, specificity exceeded 97% and sensitivity ranged from 81% to 95%. To supplement the HIV registry, we selected an algorithm with 99.89% (95% CI: 99.76% - 100.00%)

available publicly due to provisions in our service contracts, institutional policy, and ethical requirements. In order to facilitate research, we make such data available via individual data access requests. Some BC-CfE data is not available externally due to prohibitions in service contracts with our funders or data providers. Institutional policies stipulate that all external data requests require collaboration with a BC-CfE researcher. For more information or to make a request, please contact privacy@bccfe.ca.

**Funding:** Funding was provided by the BC Centre for Excellence in HIV/AIDS and the BC Ministry of Health. The funders had no role in study design, data collection and analysis, decision to publish, or preparation of the manuscript.

**Competing interests:** I have read the journal's policy and the authors of this manuscript have the following competing interests: JSGM is the Executive Director and Physician-in-Chief of the BC Centre for Excellence in HIV/AIDS, a provincial program serving all BC health authorities, and based at St. Paul's Hospital-Providence Health Care. JM's Treatment as Prevention® (TasP®) research, paid to his institution, has received support from the BC Ministry of Health, Health Canada, Public Health Agency of Canada, Genome BC, Vancouver Coastal Health and VGH Foundation. Institutional grants have been provided by Gilead, Merck and ViiV Healthcare. JSGM received no specific funding for this work and has no competing interests. VDL is funded by a grant from the Canadian Institutes of Health Research, the Canadian Foundation for AIDS Research (CANFAR Innovation Grant – 30-101), and has received honoraria for the CROI Ambassador Program from ViiV Healthcare. KK is funded by a Michael Smith Health Research BC Research Trainee Fellowship (grant number: #RT-2022-2559), a CTN Postdoctoral Fellowship (no grant number), and a Canadian Institutes of Health Research Postdoctoral Fellowship (grant number: HIV 181935). The other authors declare that they have no competing interests. This does not alter our adherence to PLOS ONE policies on sharing data and materials.

specificity and 82.21% (95% CI: 81.26% - 83.16%) sensitivity, requiring five HIV-related healthcare practitioner encounters or two HIV-related hospitalizations within a 12-month window, or one hospitalization with HIV as the most responsible diagnosis. Upon adding PWH identified by this highly-specific algorithm to the registry, 8,774 PWH were present in BC as of March 2020, of whom 333 (3.8%) were algorithm-identified.

## Discussion

In the context of an existing low prevalence disease registry, the results of our validation study demonstrate the value of highly-specific case-finding algorithms applied to administrative healthcare records to enhance our ability to estimate the number of PWH living in BC.

## Introduction

Administrative healthcare records obtained primarily for financial and budgeting purposes have become an invaluable data source for secondary use, such as disease monitoring, research, and health service planning [1, 2]. However, billing errors, miscoded diagnoses, and other data quality concerns can lead to misclassification–and hence biased findings/inferences–if such records are assumed to be valid for scientific research in their raw form [3, 4]. For instance, assuming a person has a condition identified based on a single healthcare practitioner encounter labeled with a relevant International Classification of Diseases (ICD) diagnostic code is generally not recommended as this single occurrence may be subject to misdiagnosis, data entry error, or be provisional/unconfirmed [3, 5, 6]. Case-finding algorithms (alternatively termed: case definitions [7], case ascertainment algorithms [8], or computable phenotypes [9]) can be applied to reduce misclassification. These algorithms assume that if a person exhibits a particular healthcare use pattern (e.g., two healthcare practitioner encounters labeled with particular ICD diagnostic codes within 12 months), they may be more validly ascertained as a case.

Case-finding algorithms are typically evaluated using a validation sub-sample of persons with healthcare records linked to a reference standard, which is a reliable indicator of disease status (e.g., diagnostic test, chart review). A measurement tool (e.g., a case-finding algorithm) can be applied to healthcare records, and results can then be compared with the reference standard [3]. Validity evidence derived from analyzing this validation sub-sample is assumed to be generalizable to settings where only the algorithm can be applied and the reference standard is unavailable. Common validation metrics include a) sensitivity, which indicates the ability of an algorithm to correctly identify a person as having a disease (a 'case'); the fewer the false negatives, the higher the sensitivity, and b) specificity, which indicates the ability of an algorithm to correctly identify a person as not having the disease (a 'non-case'); the fewer the false positives, the higher the specificity.

For low prevalence diseases, near-perfect specificity is particularly important since minor decreases in specificity can introduce many false positives [5, 10]. Consider an algorithm created to identify people with a low prevalence disease (0.5% prevalence). In a general population sample of 1 million people, one expects there to be 5,000 cases with that low prevalence disease (0.5% of 1 million) and 995,000 non-cases. When applying an algorithm to the sample, for each 1% decrease in algorithm specificity, almost 10,000 false positives would be generated (1% of the 995,000 non-cases) [5]. Conversely, each 1% decrease in sensitivity generates only 50 false negatives (1% of the 5,000 'true' cases) [5]. Thus, a highly-specific algorithm is crucial

to mitigate substantial misclassification bias and over-counting of persons living with a low prevalence disease.

Considering such challenges related to misclassification of disease status via algorithms, established disease registries (i.e., collections of confirmed cases ascertained using highly accurate indicators) are often preferred for case-finding and monitoring/surveillance for low prevalence diseases. Registries, however, do also have important limitations, such as incomplete capture of cases through challenges in identifying/enrolling persons, and the introduction of selection bias whereby enrollees in registries can differ from those with the disease but missing from the registry [11]. Thus, a hybrid approach of supplementing an existing low prevalence disease registry with a highly-specific case-finding algorithm (applied to healthcare records) could more validly classify a person as having or not having the disease, leading to more accurate estimates of disease burden, and can help support public health programming efforts.

## HIV in British Columbia: A case study of a low prevalence disease registry

In Canada, there were an estimated 62,790 people with HIV (PWH) as of 2020 (0.17% of the population), of which 75.4% were men. In British Columbia (BC), the prevalence of HIV was 0.19% (n = 9,637 PWH), of which approximately 82.4% were men [12, 13]. As of 2020, in terms of the main key population groups, approximately half of PWH (50.3%) in Canada were gay, bisexual, and other men who have sex with men (gbMSM), approximately one third (32.8%) were heterosexual, and 13.3% were people who inject/injected drugs (PWID) [12]. In BC (2020), among PWH, 53.8% were gbMSM, 25.0% were heterosexual, and 17.4% were PWID. The HIV incidence in 2020 was 4.0 per 100,000 population in Canada (n = 1,520 newly-identified PWH) and 2.1 per 100,000 population in BC (n = 108) [12].

The number of PWH in BC can be estimated, primarily, by examining antiretroviral treatment dispensations as recorded in the BC Centre for Excellence in HIV/AIDS (BC-CfE) Drug Treatment Program (DTP) [14]. Funded by the BC Ministry of Health (MoH), the DTP provides fully subsidized medications for the treatment of HIV (antiretroviral therapy: ART) and HIV prevention (pre-exposure prophylaxis: PrEP, and post-exposure prophylaxis: PEP) to medically eligible individuals in BC at no direct cost to program participants through a centralized program [14]. Such records represent virtually all coverage of publicly-funded ART for HIV in BC. Additionally, the DTP includes information on HIV plasma viral loads (pVL; included in the DTP via a linkage with the dedicated laboratory) [14–16]. Thus, DTP enrolment captures people with known/documented HIV positivity based on detectable pVLs, and/or ART.

In addition to PWH identified via the DTP, a relatively small number of persons may be identified via other non-DTP sources. In particular, persons not enrolled in the DTP but with a positive HIV test reported to the BC Centre for Disease Control (BCCDC), or persons whose death is coded as HIV/AIDS-related by the BC Vital Statistics Agency [16, 17]. These non-DTP indications of HIV positivity are linked to DTP records as part of the Seek and Treat for Optimal Prevention of HIV/AIDS (STOP HIV/AIDS) program evaluation data linkage [18]. This is a population-based linkage of healthcare, clinical, and HIV testing records for PWH in BC, established to monitor HIV related outcomes of the STOP HIV/AIDS Program. Hence, we refer hereafter to the HIV 'registry' as those PWH in BC identified via a detectable pVL, ART dispensation (BC-CfE DTP), a positive HIV test (BCCDC), and/or an HIV/AIDS cause of death (BC Vital Statistics).

Some PWH, however, are not present in this registry due to lacking documented HIV status based on the noted criteria. For instance, PWH may be missing from the BC-CfE's DTP records if receiving ART from a source other than the DTP; these sources include private

insurance, community pharmacies where one can pay out-of-pocket, non-DTP clinical trials or studies, and federal programs such as the Interim Federal Health, Military, and Royal Canadian Mounted Police plans [19] or Health Canada's Non-Insured Health Benefits (NIHB) for First Nations and Inuit Health [20]. BC residents who have coverage under federal programs such as NIHB or Interim Federal Health, historically and currently, have been able to access ART through the BC-CfE's DTP, or another insurance plan outside the BC-CfE at a community pharmacy. In 2017, the provision of pharmacy services and prescription drugs for most First Nations people living in BC was transitioned from NIHB to BC MoH's PharmaCare plan, including the transition of specialty pharmacy services for antiretroviral medication from NIHB to BC-CfE, the responsible provincial agency [21]. However, a small number of persons were not transitioned to the BC MoH's PharmaCare plan (within the BC-CfE) and hence continued to be able to access ART via a community pharmacy instead. For individuals receiving medications outside the BC-CfE, ART records at the BC-CfE may be absent, and hence are missing from the registry. Therefore, to help facilitate identification of such persons missing from the registry, case-finding algorithms can be applied to BC healthcare records. It must be noted that any PWH not present in either the HIV registry or without any HIV-related healthcare use in BC will de facto not be captured by any healthcare use-based algorithm.

## Prior HIV case-finding algorithms in Canada

Several HIV case-finding algorithms for administrative data have been published [22–24], including three in Canada [5, 25, 26]. Since healthcare systems, healthcare use patterns, and diagnostic coding vary by jurisdiction [27–30], an algorithm validated in Canada (ideally, in BC) is preferable. An Ontario-based algorithm identified PWH via three HIV-related physician encounters occurring within 36 months [5]. This algorithm was validated using a subsample of clients in Ontario rostered to physicians in two urban, Toronto-based clinics between 2005–2008. In these clinics, PWH represented ~20% of all clients, a proportion much higher than in the general population, which may challenge the generalizability to a province-wide, population-based sample of persons with diverse levels of healthcare engagement in BC. With a goal of minimizing false positives, a Manitoba-based algorithm identified PWH via six HIV-related physician encounters occurring within 24 months; the authors presented additional goal-specific algorithms as well [26]. In this study, algorithms based on physician encounters, hospitalizations, and medication dispensations occurring from 2007 to 2018 were validated against positive HIV tests, using virtually the entire provincial population (present during the study period) as the validation sub-sample. Although comprehensive, the Manitoba-based algorithm study aimed at providing multiple options for algorithms rather than targeting a highly-specific algorithm as the sole focus per se (the scenario for the present analysis). Furthermore, a BC-based previously study identified PWH via three HIV-related physician encounters or one HIV-related hospitalization at any time [25]. However, specificity was not calculated as records of negative HIV tests were unavailable.

We are re-examining the BC-based HIV case-finding algorithm for several reasons. The latest year of data for earlier work [25] was 2010, representing a key time point in the landscape of HIV healthcare provision in BC. This period aligned with the initiation of the STOP HIV/AIDS pilot program, which was expanded to the rest of the province in 2013 [18]. One of the program's aims was to increase HIV testing, which would be expected to yield additional previously undiagnosed HIV infections and likely lead to higher volumes of HIV-related healthcare use. A by-product of increased HIV-related healthcare use would be a concordant increase in erroneous HIV-related billings. For instance, billings by healthcare practitioners that were not for active care/treatment of PWH but likely related to discussing, for example, a

negative HIV test result, risk factors related to HIV, PEP after a high-risk exposure, or a PrEP prescription (which became available as a provincially-funded benefit on January 1st, 2018 [31]), would result in 'HIV-related visits' for persons without HIV. Another reason for re-examining an algorithm is to examine the role of specificity and the tangible implications of false positives; specificity was not computed in the previous BC-based algorithm, yet is a crucial criterion on which to select an algorithm when supplementing a low prevalence disease registry [25].

We aimed to demonstrate the feasibility of using an algorithm with high specificity for a low prevalence disease and to supplement the count of PWH previously identified via the HIV registry within the STOP HIV/AIDS linkage (i.e., a detectable pVL, ART, positive HIV test, or HIV/AIDS-related death). We addressed these aims by creating and validating a comprehensive set of case-finding algorithms to identify PWH in BC, Canada.

## Methods

### Validation sub-sample

A validation sub-sample was derived from persons in the STOP HIV/AIDS data linkage–a population-based linkage of healthcare, clinical, and HIV testing data for PWH in BC [14–17, 32–35]. The BC MoH provided healthcare records (healthcare practitioner encounters, hospitalizations, mortality records, medication dispensations), the BCCDC provided HIV test results, and the BC-CfE provided HIV-related clinical information (e.g., ART dispensations, pVLs). Data linkage was completed in 2022, and analyses were performed from January to March 2023. HIV tests were the reference standard compared against each proposed case-finding algorithm based on healthcare records. We focused on HIV tests and healthcare records between April 1996 and March 2020 (i.e., the STOP HIV/AIDS data linkage study period). Among all persons in the linkage (n = 15,957), 7,124 had a valid positive serologic HIV test, of which 2,817 persons additionally contributed a valid negative HIV test result occurring before their valid positive test (Fig 1).

### The reference standard: HIV test results

HIV is a reportable infection and all positive HIV test results completed in the province are reported to the BCCDC. The BCCDC provided all HIV diagnoses (new and previous) as recorded in the HIV/AIDS surveillance system [16]. These HIV diagnoses were linked to HIV tests performed by the BCCDC Public Health Laboratory, which conducts >95% of all HIV screening tests province-wide, and all confirmatory testing. Rapid, point-of-care tests are not included; however, all positive point-of-care tests undergo confirmatory testing before being considered a new HIV diagnosis [36]. All positive HIV test results are followed up by public health nurses to determine if it is a new HIV diagnosis or if it may be a false positive. False positive test results are flagged as not being a new diagnosis in the provincial HIV/AIDS surveillance system. If a new HIV diagnosis is identified, the public health nurses obtain information about risk factors for HIV acquisition and provide partner notification services and referrals for HIV care. (See the Supporting Information File for further HIV test-related details.)

In the HIV testing data extract available within the STOP HIV/AIDS data linkage, negative HIV test results were available only for persons with at least one positive HIV test on record with the BCCDC (i.e., all persons in the validation subsample had a positive HIV test, but a subset had a least one prior negative HIV test[s] recorded, as well). As an additional quality assurance step, negative HIV test results that occurred after indications of HIV positivity (an ART medication dispensation or detectable pVL) were omitted (Fig 1). If a person had multiple negative HIV tests, the final negative HIV test before the positive HIV test was retained.

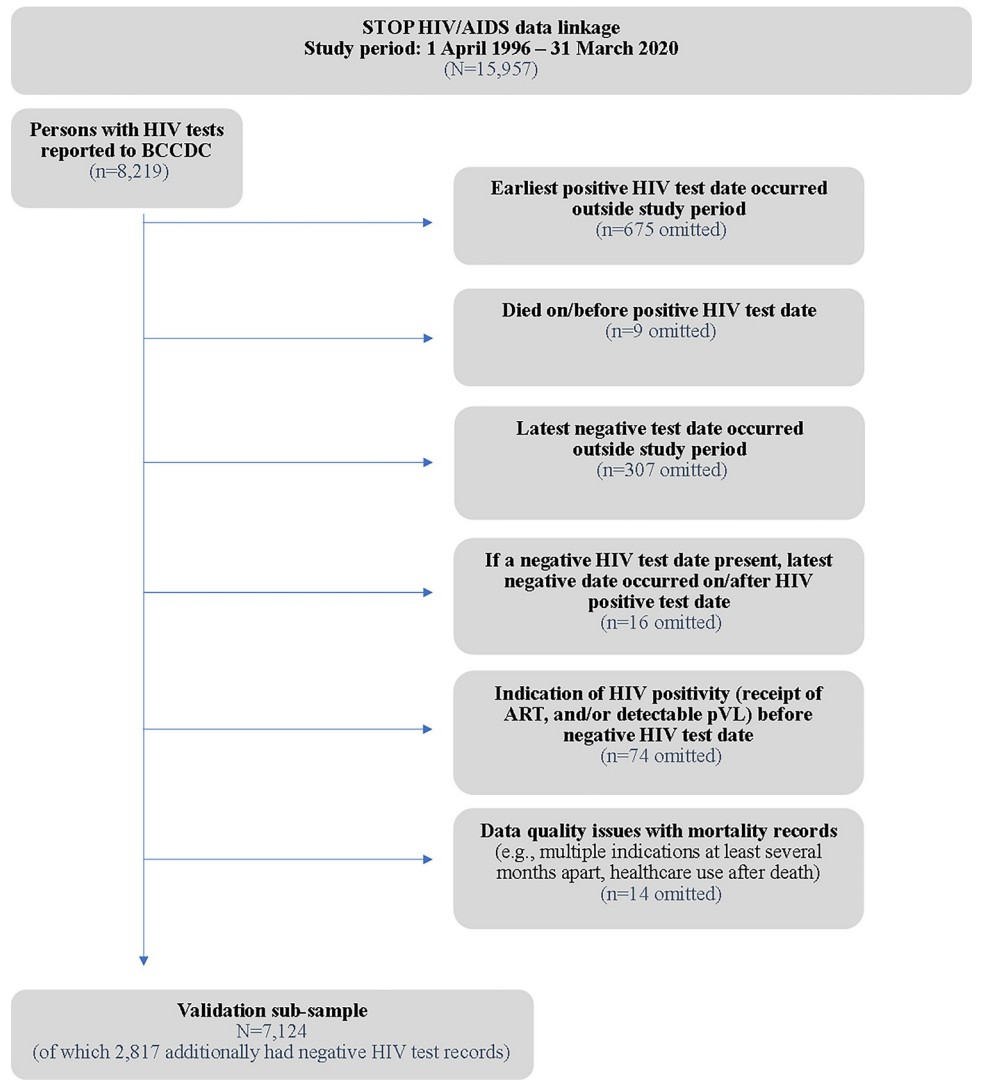

**Fig 1. Creation of the validation sub-sample to evaluate HIV case-finding algorithms.** ART: Antiretroviral therapy medication dispensations. pVL: plasma viral load for HIV.

This reference standard information was used to compartmentalize a participant's person-time to identify when someone had HIV (i.e., the period of all available healthcare records following their positive HIV test date) or not (i.e., the period of all available healthcare records preceding their latest negative HIV test date). See the Supporting Information File for visual examples of these concepts.

## The comparator: Case-finding algorithms based on healthcare use

Case-finding algorithms were based on combinations of HIV-related healthcare documented in healthcare practitioner and hospitalization records. The BC MoH provided the Medical Services Plan (MSP) Payment Information File [32], containing healthcare practitioner encounter billings (including, but not limited to, billings for services by physicians and nurse practitioners), and the Discharge Abstract Database (DAD) containing data on hospitalizations and day surgeries in BC [34]. Published Canadian case-finding algorithms with hospitalizations as a

criterion usually require a single hospitalization, because data entry in the DAD is standardized, with trained abstractors and quality assurance activities in place [8, 25, 37, 38]. (Conversely, data entry in MSP billings has less quality assurance: typically, the practitioner or medical office assistant enters the information). Over time, however, hospital re-abstraction studies have found a non-trivial level of discordance between diagnoses recorded in the DAD hospital records versus the original chart [39, 40]. Such discordance was particularly evident for diagnoses not recorded in the DAD's 'Most Responsible Diagnosis' (MRD) field [39–41]. Perhaps as a reflection of this, some Canadian algorithms require multiple hospitalizations in their criteria [7, 42]. Therefore, we examined both single and multiple hospitalizations criteria in our algorithms, with the assumption that multiple hospitalizations with particular diagnostic codes (e.g., HIV-related) would corroborate evidence against potential coding errors.

In our study, each case-finding algorithm was constructed based on combinations of healthcare practitioner encounters and hospitalization records. An HIV-related healthcare practitioner encounter was defined as an encounter by a unique client (patient) with a unique healthcare practitioner, and on a unique service date [43, 44]. This definition was used since a unique healthcare practitioner encounter may contain multiple MSP fee items (i.e., distinct activities/services provided within an encounter, causing the single encounter to be split into multiple records in the MSP file), and to ensure same-day encounters by one client to multiple unique healthcare practitioners would be counted as separate, unique encounters. An HIV-related hospitalization was defined as a single hospitalization record using the discharge date. Healthcare practitioner encounters and hospitalizations were defined as HIV-related if they had a relevant ICD diagnostic code associated with the visit or hospitalization, indicating that HIV-related care was provided and/or that the client was likely a PWH (see the Supporting Information File for details). Aligned with previous Canadian case-finding algorithm validation studies [5, 7], hospitalization records, rather than hospitalization episodes of care (counting sequential, transfer-related hospitalization records into single, distinct events) [45], were the units of analysis for enumerating hospitalizations.

## Algorithm validation

To evaluate how well algorithms correctly classified HIV cases based on the reference standard (lab-confirmed positive HIV status), we estimated the sensitivity and specificity of all proposed algorithms. Each algorithm reflected a unique healthcare use pattern based on HIV-related healthcare practitioner encounters or hospitalizations. We examined a range of healthcare use patterns using combinations of 1–2 hospitalizations or 1–5 healthcare practitioner encounters, occurring within four search windows (12, 24, 36 months) for a total of 28 unique algorithms.

To estimate sensitivity, healthcare records in the period from the positive HIV test result date until the end of follow-up were searched to identify the HIV-related healthcare use pattern for each of the proposed algorithms (e.g., two HIV-related healthcare practitioner encounters within 12 months). Sensitivity was the proportion of people who correctly met the algorithm after the positive HIV test date until the end of follow-up out of all lab-confirmed PWH. To estimate specificity, the period before the negative HIV test date until the start of follow-up was searched to identify the absence of that same HIV-related healthcare pattern defined by the algorithm. The sensitivity and specificity of algorithms were stratified by sex, age at positive HIV test date, and key population groups [3].

Search windows (i.e., the period within which the HIV-related healthcare pattern of a given algorithm had to co-occur) varied from 12 to 36 months, and for single events (i.e., one healthcare practitioner encounter or hospitalization), no explicit window was required. To ensure an appropriate denominator, namely, to ensure persons were present in BC for a minimum

amount of time to be eligible to access some healthcare, and to omit persons who left BC or died recently after a positive HIV test, criteria were applied. For sensitivity estimates, persons needed to be present in BC for at least 12 months after their positive HIV test, and for specificity estimates, persons with a negative HIV test needed to be present in BC for at least 12 months before their negative HIV test.

Although specificity was the primary metric of interest when evaluating the algorithms, for completeness, we reported two composite metrics that equally weighted sensitivity and specificity (see the Supporting Information File for a description of the metrics and the values per algorithm): 1) The area under the curve (AUC)–also known as the Concordance-/C-statistic [46], and 2) the Kappa statistic [47]. Using the Wald method, we computed point estimates and 95% confidence intervals (95% CI) for all four metrics [48]. Data were analyzed using SAS 9.4 and visualized using R 4.2.2 [49, 50].

After assessing algorithms within the validation sub-sample, we applied algorithms to a 'candidate pool' of possible PWH within the STOP HIV/AIDS data linkage. This pool contains healthcare records of any persons to have recorded any HIV-related healthcare use (healthcare practitioner encounters or hospitalizations) between April 1996 and March 2020. This step demonstrated how the total BC-wide estimate of PWH, present as of March 31, 2020, varied after supplementing the count of PWH identified via indications of HIV positivity within records in the HIV registry with those identified via each case-finding algorithm. Persons were identified as PWH if they a) were identified based on records in the registry part of the STOP HIV/AIDS data linkage indicating HIV positivity (as described previously), or b) met the criteria of the case-finding algorithm. To assess algorithms, we examined such patterns in counts of PWH and performance (primarily specificity) of algorithms applied in the validation sub-sample.

## Ethics

Linkage and use of datasets were approved and performed by data stewards in each collaborating agency, facilitated by the BC Ministry of Health. The University of British Columbia/Providence Health Care Ethics Review Board at the St. Paul's Hospital site granted ethics approval for this study (H18-02208). This study was conducted using strictly anonymized laboratory, clinical, and healthcare records; thus, informed consent was not required. This study complies with BC's Freedom of Information and Protection of Privacy Act.

## Results

In the validation sub-sample, the median age at the positive HIV test date was 37 years (Q1: 30, Q3: 46), 48.9% were residents of Vancouver Coastal Health Authority (a region encompassing Vancouver, BC's largest municipality), and 80.1% were men. Information on key population group was sourced from BC-CfE records (DTP enrolment form or historical information from surveys), BCCDC HIV testing records (the risk factors and exposure information field in the HIV case report form), and for those identified as PWID, PharmaNet medication dispensations were additionally used (medications for opioid agonist therapy, such as methadone). The largest group (36.7%) included those who identified as gbMSM, 29.6% who were ever PWID, 7.9% identified both as gbMSM and PWID, 18.0% identified as heterosexual, 1.9% reported another HIV transmission risk, and for 6.0% this information was missing. The median annual rate of MSP-billed, all-cause outpatient healthcare practitioner encounters for persons in the validation sub-sample was 9.7 encounters/year. See the Supporting Information File for additional information regarding characteristics of the validation sub-sample, characteristics of those in the STOP HIV/AIDS data linkage who were excluded from, and

comparisons of those within the validation sub-sample who had solely a positive HIV test record versus those with both a positive and a negative HIV test record.

A median of 9.0 years of follow-up time were available when querying for HIV-related healthcare records before the negative HIV test result date (i.e., to estimate specificity). A median of 8.1 years of healthcare records were available when querying for HIV-related healthcare records after the positive HIV test result (i.e., to estimate sensitivity). Detailed results of all sensitivity and specificity estimates per algorithm, stratified by search window, are presented in the Supporting Information File.

## Specificity

Specificity ranged from 97.13% (95% CI: 96.50% - 97.76%; algorithm requiring one HIV-related healthcare practitioner encounter or one hospitalization, ever) to 99.89% (95% CI: 99.76% - 100.00%; algorithm requiring five healthcare practitioner encounters or two hospitalizations within a 12-month window; see Fig 2). The most specific algorithms required several HIV-related healthcare practitioner encounters and/or several hospitalizations, whereas the least specific algorithms required a single HIV-related healthcare practitioner encounter. Search window length had a negligible impact on specificity.

## Sensitivity

Sensitivity ranged from 80.82% (95% CI: 79.84% - 81.80%; five HIV-related healthcare practitioner encounters or two hospitalizations within a 12-month window) to 95.47% (95% CI: 94.96% - 95.99%; one HIV-related healthcare practitioner encounter or one hospitalization ever; see Fig 3). The most sensitive algorithms tended to require one HIV-related healthcare practitioner encounter. The least sensitive algorithms required several HIV-related healthcare practitioner encounters or hospitalizations. Longer search windows led to increased sensitivity (e.g., the sensitivity of the five HIV-related healthcare practitioner or two hospitalizations algorithm increased from 80.82% [95% CI: 79.84% - 81.80%] for a 12-month window to 85.68% [95% CI: 84.81% - 86.55%] for a 36-month window).

## Strata-specific performance and additional metrics

Sensitivity and specificity were estimated for algorithms across three strata: sex (male/female; based on BC MoH records), age at first HIV-related record (over 30 years vs. 30 years or younger; based on BC MoH, and BC-CfE records), and key population group (binary indicator for PWID, gbMSM, and heterosexual status; based on BC-CfE, BCCDC HIV test records, and medication dispensing records, BCCDC HIV test records). Sensitivity tended to be slightly higher (indicated by non-overlapping 95% CIs) for: males, persons aged over 30 years, the PWID (vs. non-PWID) group, the gbMSM (vs. non-gbMSM) group, and the non-heterosexual (vs. heterosexual) group. For specificity, due to the smaller sample size of persons with negative HIV test results (a subset of the validation sample), across all stratifications, the 95% CIs were wide and overlapping between groups, thus preventing interpretation of group differences. See the Supporting Information File for all stratified performance estimates and C-statistic and Kappa values.

## Implementing algorithms when estimating the total count of PWH in BC

After estimating performance of the algorithm in the validation sub-sample, total counts of PWH alive in BC as of March 2020 were estimated by combining the count of persons identified as PWH via indications of HIV positivity in the registry (DTP or a positive HIV test

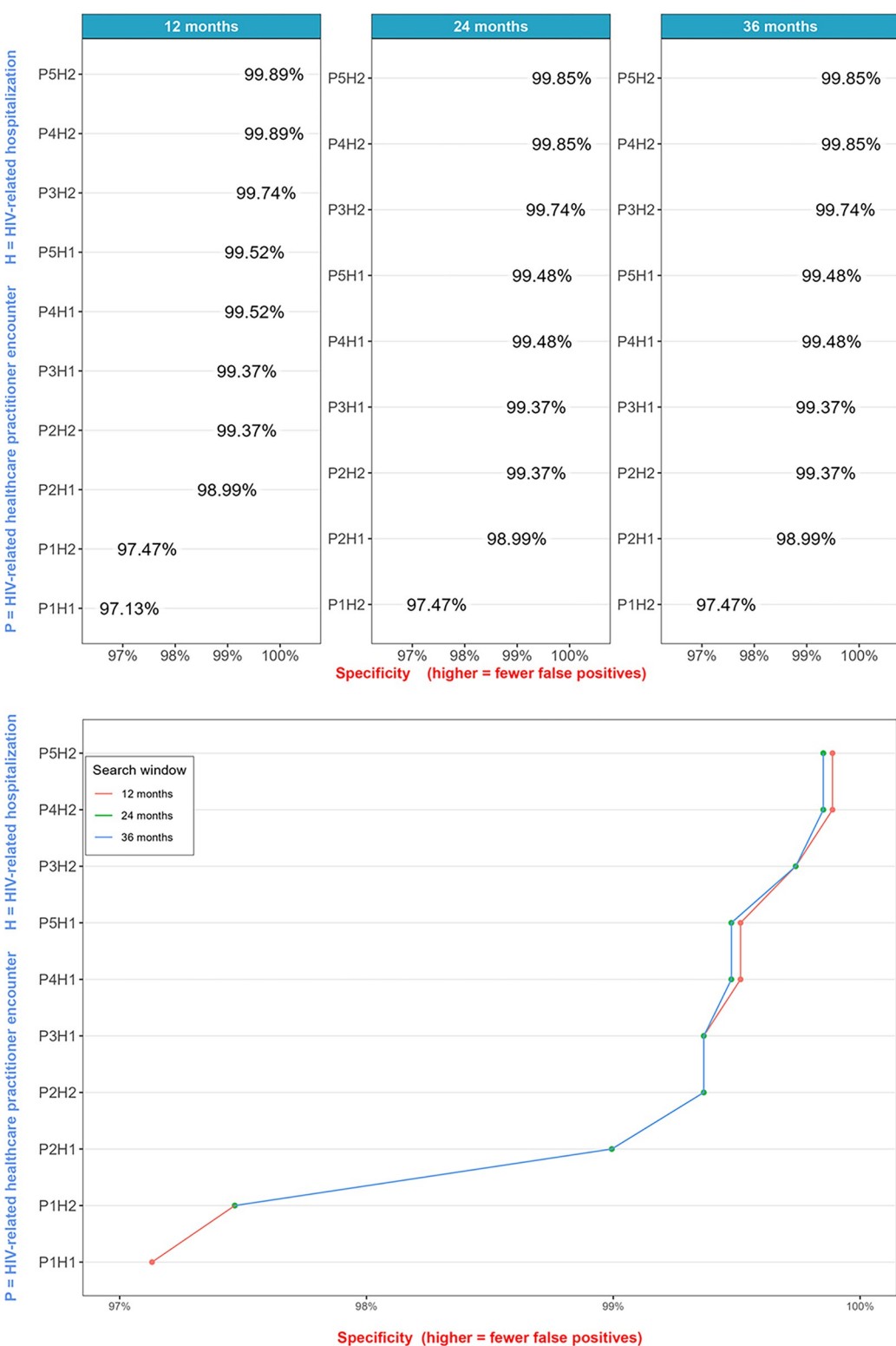

**Fig 2. Specificity estimates from the validation sub-sample, stratified by search window.** *The H1 and P1 events were unbounded by search windows since they contained single events; hence, the algorithm P1H1 referred to 1 HIV-related healthcare practitioner, or 1 HIV-related hospitalization occurring at any time during a person's follow-up.

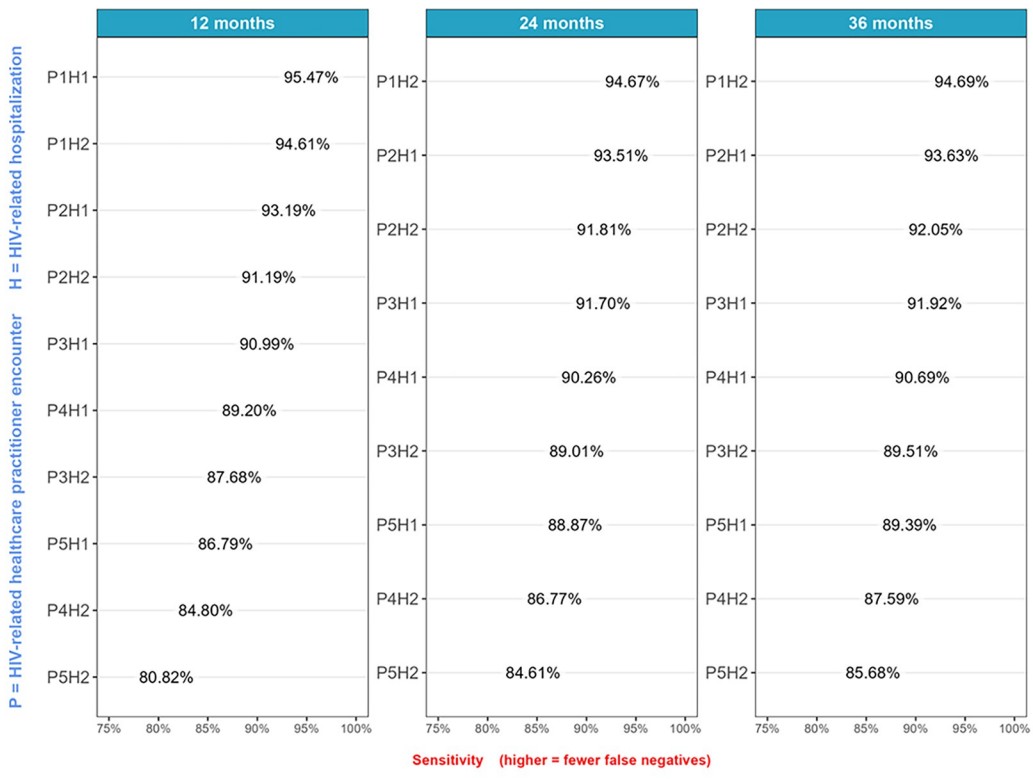

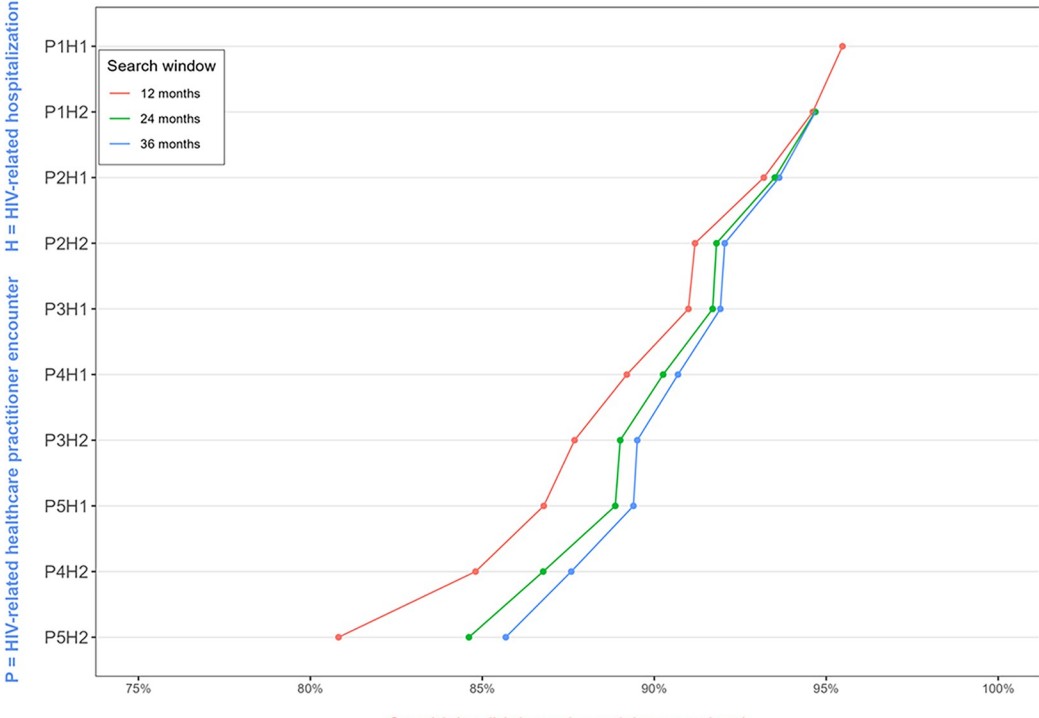

**Fig 3. Sensitivity estimates from the validation sub-sample, stratified by search window.** *The H1 and P1 events were unbounded by search windows since they contained single events; hence, the algorithm P1H1 referred to 1 HIV-related healthcare practitioner, or 1 HIV-related hospitalization occurring at any time during a person's follow-up.

reported to the BCCDC) and those identified solely via each algorithm (Fig 4). Total counts of PWH present in BC as of March 2020 varied substantially by algorithm, from 8,755 (algorithm requiring five HIV-related healthcare practitioner encounters or two hospitalizations within a 12-month window) to 30,071 (algorithm requiring one HIV-related healthcare practitioner encounter or one hospitalization, ever). An expected pattern emerged whereby higher algorithm specificity corresponded to smaller counts of PWH identified. See the Supporting Information File for patterns of estimated counts of PWH vs. specificity, and overall PWH counts by algorithm.

## Final evaluation and selection

Two algorithms were retained for further consideration, based on their high specificity and lower estimated algorithm-identified counts of PWH (when supplementing the count of PWH sourced from indications of positivity in the HIV registry). P5H2Mon12 (i.e., five HIV-related healthcare practitioner encounters OR two hospitalizations occurring within a 12-month window) was retained since it yielded the highest specificity and the smallest number of algorithm-identified PWH. This algorithm was assumed to be most effective at minimizing false positives. P5H1Mon12 (i.e., five HIV-related healthcare practitioner encounters occurring within a 12-month window OR one HIV-related hospitalization ever) was retained as it mirrored the same healthcare pattern, except it required a single hospitalization, thus serving as a useful comparison. Both algorithms exhibited very high specificity (P5H2Mon12: 99.89% [95% CI: 99.76% - 100.00%]; P5H1Mon12: 99.52% [95% CI: 99.25% - 99.78%]) and reasonable sensitivity (P5H2Mon12: 80.82% [95% CI: 79.84% - 81.80%]; P5H1Mon12: 86.79% [95% CI: 85.95% - 87.63%]).

The hospitalizations criterion of the P5H2Mon12 algorithm was expanded to allow case identification via one hospitalization only if an HIV-related diagnostic code was listed as the MRD for the hospital record. This modification was informed by evidence of some data quality concerns with hospitalization records in the DAD from hospital re-abstraction studies, particularly for non-MRD diagnoses. As indicated in Table 1, this modified algorithm demonstrated 99.89% (95% CI: 99.76% - 100.00%) specificity and 82.21% (95% CI: 81.26% - 83.16%) sensitivity, thus exhibiting specificity equal to the P5H2Mon12 algorithm and higher than the P5H1Mon12 algorithm, and sensitivity higher than the P5H2Mon12 algorithm. Applying this modified algorithm identified 333 additional PWH otherwise not identified via the registry. See the Supporting Information File for further characteristics of the total PWH in BC, by method of identification (algorithm vs. non-algorithm).

## Discussion

This study demonstrated the value of supplementing a low prevalence disease registry with additional cases identified by applying a highly specific case-finding algorithm. We evaluated 28 case-finding algorithms for HIV and found that an algorithm requiring five HIV-related healthcare practitioner encounters or two HIV-related hospitalizations within a 12-month window, or one hospitalization with HIV as the MRD had the highest specificity (99.89%) while maintaining reasonable sensitivity (exceeding 82%). This algorithm led to the identification of over 300 additional PWH present in BC as of March 2020, previously not identified via any indication of HIV positivity via the HIV registry (detectable pVL, ART, positive HIV test, or HIV/AIDS-related death). In a setting where a disease registry is unavailable, or is less robust, one may consider an algorithm requiring five HIV-related healthcare practitioner encounters within a 12-month window, or one HIV-related hospitalization ever (P5H1Mon12). This algorithm maintains high specificity–albeit not as high as our chosen

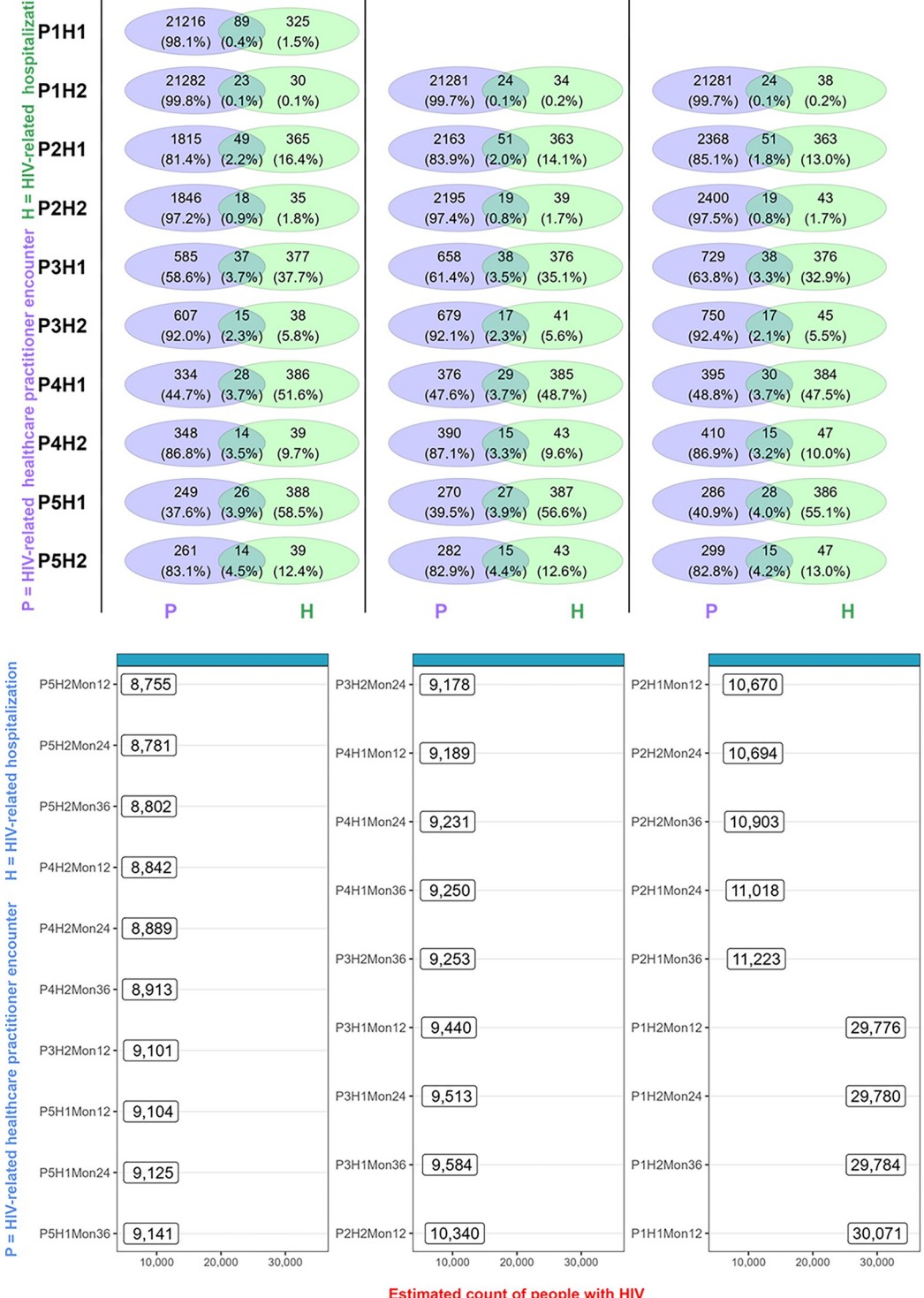

**Fig 4. Composition of algorithm-identified persons with HIV by healthcare source, and implementation of algorithms to supplement the existing registry to estimate total counts of persons with HIV alive as of March 2020 in BC, Canada.** * Top figure visualizes the composition of the algorithm-identified persons (identified via healthcare practitioner encounters, hospitalizations, or both); bottom figure indicates the estimated counts of persons with HIV alive in BC based on using each algorithm-identified count to supplement the registry-sourced count of persons with HIV. The

H1 and P1 events were unbounded by search windows since they contained single events; hence, the algorithm P1H1 referred to 1 HIV-related healthcare practitioner, or 1 HIV-related hospitalization occurring at any time during a person's follow-up.

algorithm–but has slightly higher sensitivity (hence, a higher composite score than the P5H2Mon12 algorithm, as indicated by Kappa and C-statistic values; see the Supporting Information File). This may be a preferable trade-off in the absence of a robust low prevalence disease registry, where higher sensitivity would be more important. Nevertheless, validation work should be conducted in other provinces/jurisdictions to understand the extent to which such an algorithm may be applicable to those settings.

The previously published BC-based HIV algorithm [25] demonstrated a sensitivity of 88.0% (specificity was not computed for this algorithm). It required three HIV-related healthcare practitioner encounters or one HIV-related hospitalization without any search window within which the healthcare practitioner encounters had to co-occur. The lack of a search window provides more opportunity for false positives (lower specificity) since several healthcare encounters with billing errors could foreseeably accumulate over the expanding years of follow-up. The most comparable algorithm to the previous BC-based algorithm in the present study (three HIV-related healthcare practitioner encounters within 36 months, or one HIV-related hospitalization) demonstrated a sensitivity of 91.92%, slightly higher than the 88.0% reported by the previous BC-based algorithm [25]. The slightly higher level of sensitivity observed in the present study may be partly attributed to the longer follow-up length available for persons to record HIV-related healthcare use (January 1995 to March 2010 vs. March 1996 to March 2020, in the present study), providing a longer period in which to observe the required healthcare use within a search window.

## Conclusions and implications

In conclusion, our study identified an algorithm to be highly-specific in identifying PWH against a reference standard, and when applied to an existing registry of PWH, helped supplement the count to yield a more accurate overall prevalence estimate. This exercise underscored the value of supplementing a low prevalence disease registry with an algorithm applied to healthcare records. Another such example was demonstrated for pediatric pulmonary hypertension [9], where they supplemented their registry with a case-finding algorithm from electronic health records. Application of their algorithms to these health records helped identify several hundred additional cases otherwise not identified in their registry.

In an applied sense, our findings can enhance the ability to identify PWH in BC. By extension, having more reliable estimates of PWH can help ensure interventions (such as support

**Table 1. Performance of the three final candidate HIV case-finding algorithms.**

| Algorithm | Specificity (95% CI) | Sensitivity (95% CI) | Algorithm-identified HIV cases (Total PWH*) |
|---|---|---|---|
| P5H2Mon12 | **99.89%** (99.76% - 100.00%) | **80.82%** (79.84% - 81.80%) | 314 (8,755) |
| P5H2Mon12H1MRD (modified) | **99.89%** (99.76% - 100.00%) | **82.21%** (81.26% - 83.16%) | 333 (8,774) |
| P5H1Mon12 | **99.52%** (99.25% - 99.78%) | **86.79%** (85.95% - 87.63%) | 663 (9,104) |

*Total persons with HIV (PWH) as of March 2020; based on sum of cases identified via information in the STOP HIV/AIDS data linkage (antiretroviral medications, detectable viral loads, a positive HIV test reported to the BCCDC) OR via the case-finding algorithm. CI: Confidence interval. P: HIV-related healthcare practitioner encounter. H: HIV-related hospitalization. MRD: HIV-related hospitalization where HIV was the most responsible diagnosis. The H1 and P1 events were unbounded by search windows since they contained single events; hence, the algorithm P1H1 referred to 1 HIV-related healthcare practitioner, or 1 HIV-related hospitalization occurring at any time during a person's follow-up.

services) are more appropriately targeted and scaled. Additionally, it can also improve accuracy of public health surveillance and can support overall public health prevention efforts [51].

## Strengths and limitations

Study limitations exist when using case-finding algorithms based on healthcare records. Sensitivity will be better for persons with frequent documented healthcare use than for those with little/no documented healthcare use. Thus, PWH in BC without any documented HIV-related healthcare use will be missing from the algorithm-contributed portion of estimates of PWH. Conversely, high-level healthcare users may be more likely to receive false positive classifications by virtue of registering more encounters–thus attenuating specificity. We note, however, that this shortcoming is partly mitigated by the comprehensive capture of PWH via HIV registry–permitting us to prioritize specificity. Another challenge was the validation sub-sample was composed of persons who all eventually tested positive for HIV in the province of BC (i.e., negative HIV test records were only available for those who had a positive HIV test on record). Therefore, the validation sub-sample without the disease (without HIV; non-cases), is sourced from the population acquired the disease at a later date. Hence, the sample without HIV may not be representative of those without HIV in the general population, and hence the algorithm's performance (particularly, specificity) may be less generalizable than would be the case if a representative sample of the general population was used for the source of non-cases. Access to all negative HIV tests, including for individuals who never went on to test positive for HIV, would have provided a better representation of healthcare use in the period before a negative HIV test result (in our validation sub-sample). Relatedly, because of the nature of the validation sub-sample, we were unable to estimate two other common validity metrics: positive and negative predictive values. To be meaningfully interpreted, these predictive values require a disease prevalence in the validation sub-sample comparable to that of the population where one intends to apply such algorithms (e.g., in a random general population sample)–an assumption not met in this study [3]. Thus, aligned with a similar study that also did not meet this assumption [5], predictive values were not estimated in the present study.

With respect to strata-specific estimates/comparisons of sensitivity and specificity, some potential differential sensitivity was observed across the subgroups (e.g., higher sensitivity for males than females). However, the metric of focus in the present study–specificity–could not be reliably compared between subgroups due to the smaller sample size of persons with negative HIV tests. Future work, with a larger number of persons with negative HIV tests, or, more generally, people without HIV, may be able to examine such subgroup differences in specificity more precisely. No stratifications were performed on factors related to ethno-cultural background as such information was unavailable in our datasets. Further, although we did not compare sensitivity/specificity by calendar era, it is shown in the Supporting Information File that the proportion of algorithm-contributed PWH was smaller among PWH present in 2020 (2.1%; 333/8,774) versus among PWH ever present in the study period (6.2%; 985/15,957). Thus, in recent years, a larger proportion of people are identified as being PWH via the HIV registry–likely in part due to enhanced testing rates in BC, and better connections to HIV care, over time. ART dispensations and detectable pVLs, are part of the HIV registry (an indicator of HIV positivity) and since we used the algorithms to supplement such indications, we focused on HIV-related healthcare use. Also, in BC, ART dispensations for HIV were unavailable in the standard provincial medication dispensation database (PharmaNet) for our study period (data up to March 2020).

Strengths of the study include the large validation sub-sample of approximately 7,100 persons; all had a positive HIV test on record in BC and approximately 2,800 had at least one

negative HIV test on record. We also considered various nuances within the many algorithms examined, including varying the numbers of HIV-related healthcare practitioner encounters and hospitalizations, and the windows within which these events occurred. Moreover, this work extended and updated the previous BC-based HIV case-finding algorithm validation study [25] by estimating specificity. Finally, we were able to draw from province-wide HIV-related healthcare records to assess validity evidence supporting our algorithms. In conclusion, the results of our study demonstrate the value of applying highly-specific case-finding algorithms to administrative healthcare records to enhance the ability to estimate the total number of PWH in BC in the context of an existing HIV disease registry.

## Supporting information

**S1 File.**
(DOCX)

## Acknowledgments

The authors would like to thank the BC Centre for Excellence in HIV/AIDS, BC Ministry of Health, BC Vital Statistics Agency, the BC Centre for Disease Control, and the institutional data stewards for granting access to the data. We also thank Dr. Mel Krajden (BCCDC) for their insight regarding HIV testing in BC.

**Disclaimer:** All inferences, opinions, and conclusions drawn in this publication are those of the author(s), and do not reflect the opinions or policies of the Data Steward(s). The BC-CfE is prohibited from making individual level data available publicly due to provisions in our service contracts, institutional policy, and ethical requirements. In order to facilitate research, we make such data available via individual data access requests. Some BC-CfE data is not available externally due to prohibitions in service contracts with our funders or data providers. Institutional policies stipulate that all external data requests require collaboration with a BC-CfE researcher. For more information or to make a request, please contact privacy@bccfe.ca.

## Author Contributions

**Conceptualization:** Scott D. Emerson, Taylor McLinden, Paul Sereda.

**Formal analysis:** Scott D. Emerson.

**Investigation:** Scott D. Emerson.

**Methodology:** Scott D. Emerson, Taylor McLinden, Paul Sereda.

**Project administration:** Taylor McLinden.

**Resources:** Julio S. G. Montaner.

**Supervision:** Taylor McLinden, Paul Sereda.

**Validation:** Scott D. Emerson.

**Visualization:** Scott D. Emerson.

**Writing – original draft:** Scott D. Emerson, Taylor McLinden.

**Writing – review & editing:** Scott D. Emerson, Taylor McLinden, Paul Sereda, Viviane D. Lima, Robert S. Hogg, Katherine W. Kooij, Amanda M. Yonkman, Kate A. Salters, David Moore, Junine Toy, Jason Wong, Theodora Consolacion, Julio S. G. Montaner, Rolando Barrios.

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
