## [Decision Letter · Decision Letter 0]

31 May 2023

PONE-D-23-08773Identification of People with Rare Diseases in Administrative Healthcare Records:

A Case Study of HIV in British Columbia, CanadaPLOS ONE

Dear Dr. Emerson,

Thank you for submitting your manuscript to PLOS ONE. After careful consideration, we feel that it has merit but does not fully meet PLOS ONE’s publication criteria as it currently stands. Therefore, we invite you to submit a revised version of the manuscript that addresses the points raised during the review process.

We look forward to receiving your revised manuscript.

Kind regards,

Edward Nicol, PhD

Academic Editor

PLOS ONE

Journal Requirements:

“I have read the journal's policy and the authors of this manuscript have the following competing interests: JSGM is the Executive Director and Physician-in-Chief of the BC Centre for Excellence in HIV/AIDS, a provincial program serving all BC health authorities, and based at St. Paul’s Hospital-Providence Health Care. JM’s Treatment as Prevention® (TasP®) research, paid to his institution, has received support from the BC Ministry of Health, Health Canada, Public Health Agency of Canada, Genome BC, Vancouver Coastal Health and VGH Foundation. Institutional grants have been provided by Gilead, Merck and ViiV Healthcare. JSGM received no specific funding for this work and has no competing interests. VDL is funded by a grant from the Canadian Institutes of Health Research, the Canadian Foundation for AIDS Research (CANFAR Innovation Grant – 30-101), and has received honoraria for the CROI Ambassador Program from ViiV Healthcare. KK is funded by a Michael Smith Health Research BC Research Trainee Fellowship (grant number: #RT-2022-2559), a CTN Postdoctoral Fellowship (no grant number), and a Canadian Institutes of Health Research Postdoctoral Fellowship (grant number: HIV 181935)”

4. Please ensure that you refer to Figure 4 in your text as, if accepted, production will need this reference to link the reader to the figure.

Reviewers' comments:

Reviewer's Responses to Questions

**Comments to the Author**

1. Is the manuscript technically sound, and do the data support the conclusions?

Reviewer #1: Yes

Reviewer #2: Yes

2. Has the statistical analysis been performed appropriately and rigorously? 

Reviewer #1: Yes

Reviewer #2: Yes

3. Have the authors made all data underlying the findings in their manuscript fully available?

Reviewer #1: No

Reviewer #2: Yes

4. Is the manuscript presented in an intelligible fashion and written in standard English?

Reviewer #1: Yes

Reviewer #2: Yes

5. Review Comments to the Author

Reviewer #1: This is a well-written manuscript describing the sensitivity and specificity of various algorithms to identify people with HIV as an adjunct to a curated database of people with HIV in British Columbia. I have several suggestions/comments and points of clarification.

1. Introduction - overall the introduction is very long and could be shortened. In particular, most readers will be aware of the tradeoffs between sensitivity and specificity, and the lengthy explanation of these terms could be abbreviated.

2. The authors describe HIV as a "rare" disease in British Columbia. More details about the HIV rates and demographics of people with HIV in Canada would be helpful.

3. The algorithms evaluated only included HIV encounters and hospitalizations, and did not take into account laboratory test results (e.g., viral load frequency) or ART prescriptions, although presumably these data were available. Please explain why these were not utilized in the algorithms investigated, and/or include additional algorithms that include these factors (e.g., see Ridgway et al. Comparison of algorithms for identifying people with HIV from electronic medical records in a large, multi-site database, JAMIA Open, Volume 5, Issue 2, July 2022).

4. The authors evaluated sensitivity and specificity of the chosen algorithms by certain demographic features (e.g., sex) but not by race. I would suggest examining the performance characteristics of the algorithms broken down by race as well, given that Black individuals are disproportionately impacted by HIV and it would be imperative to ensure that algorithms perform equally well for all racial groups.

5. The algorithms selected by the authors include the requirement for 5 HIV encounters within 12 months. This is very frequent. While PWH may have had more frequent encounters with HIV providers earlier in the study period, more recently it is not uncommon for PWH with well-controlled HIV to only see providers once or twice a year. Are the algorithms more accurate earlier in the study period? Please comment on how changes in HIV care/practice over time may impact the performance characteristics of the algorithms over time.

Reviewer #2: It is a very well written article.

The introduction is very comprehensive (on the long side); it is a very complete review of the topic of using administrative health records for building databases of specific diseases and which the criteria should be fulfilled in the design of algorithms for purposes of building the database and for the validation. This article also reviews basic concepts (In the main article and the supplement) that are relevant (I.s.: sensitivity/specificity, etc.) for the reader to understand the process.

The methodology is clearly delineated and very detailed.

The algorithms studied would be very specific for one disease process (HIV) and one geographic area (British Columbia), and as such would not be generalizable to other rare diseases or systems. However, the rigorous methodology that was employed would be a map or guide for others to follow.

Specific concerns: The sample that was used for validation of the population without the disease process (without HIV), is derived from the population that has the disease process at a later date. Although the authors make reference to this weakness in their discussion, it should be further emphasized that the sample without HIV may not be representative of that population in the general population, and hence the performance of the algorithm may vary from what the authors present (even in that specific setting).

Another concern is that this sample seems to be a high utilizer of health care (mean 9 visits), which may not be representative of other areas and may alter in important ways the performance of the algorithm.

With all the caveats, as said before, this manuscript would be a very useful resource and tool for others that would like to design administrative health records based algorithms for research or other purposes.

Minor issues:

Typos: Page 10, line 226; page 13, line 287. The first row for 12 months does not align for the 24 and 36 months. Although it is clear why and this is explained in the Tables in the appendix, it is not explained here and it can make the reader not realize it and interpret the data in a wrong way.

Figure 3 and others: The first row for 12 months(P1H1) does not align with the first row for the 24 and 36 months (P1H2). Although it is clear why and this is explained in the Tables in the appendix, it is not explained here and it can make the reader not realize it and interpret the data in a wrong way (not everyone reviews the appendix information).

6. PLOS authors have the option to publish the peer review history of their article (what does this mean?). If published, this will include your full peer review and any attached files.

Reviewer #1: No

Reviewer #2: No

---

## [Author Response · Author response to Decision Letter 0]

13 Jul 2023

Journal Requirements:

We have updated the formatting of the manuscript files accordingly.

“I have read the journal's policy and the authors of this manuscript have the following competing interests: JSGM is the Executive Director and Physician-in-Chief of the BC Centre for Excellence in HIV/AIDS, a provincial program serving all BC health authorities, and based at St. Paul’s Hospital-Providence Health Care. JM’s Treatment as Prevention® (TasP®) research, paid to his institution, has received support from the BC Ministry of Health, Health Canada, Public Health Agency of Canada, Genome BC, Vancouver Coastal Health and VGH Foundation. Institutional grants have been provided by Gilead, Merck and ViiV Healthcare. JSGM received no specific funding for this work and has no competing interests. VDL is funded by a grant from the Canadian Institutes of Health Research, the Canadian Foundation for AIDS Research (CANFAR Innovation Grant – 30-101), and has received honoraria for the CROI Ambassador Program from ViiV Healthcare. KK is funded by a Michael Smith Health Research BC Research Trainee Fellowship (grant number: #RT-2022-2559), a CTN Postdoctoral Fellowship (no grant number), and a Canadian Institutes of Health Research Postdoctoral Fellowship (grant number: HIV 181935)”

We have added this sentence to our Competing Interests statement (new text highlighted in yellow in the cover letter).

Due to legal, ethical, and privacy restrictions we unable to share individual-level data. We have updated the Data availability statement in the cover letter accordingly. (new text highlighted in yellow in the cover letter). 

4. Please ensure that you refer to Figure 4 in your text as, if accepted, production will need this reference to link the reader to the figure.

We have included a reference to figure 4 in the text.

We have included separate captions per each figure.

We have updated and included captions to the Supporting Information file.

5. Review Comments to the Author

Note that for greater clarity we have added a Venn diagram within figure 4 to visualize the means which algorithm-identified people with HIV were ascertained: by meeting the healthcare practitioners criteria, hospitalizations criteria, or both. We believe this information compliments the extant information displayed by figures and tables in the manuscript. 

Reviewer #1: 

R1.Comment1: 

Introduction - overall the introduction is very long and could be shortened. In particular, most readers will be aware of the tradeoffs between sensitivity and specificity, and the lengthy explanation of these terms could be abbreviated.

Response to R1.Comment1:

We have reduced the length of the introduction – as well as other sections throughout the manuscript - by shortening and making more concise the wording.

R1.Comment2: 

The authors describe HIV as a "rare" disease in British Columbia. More details about the HIV rates and demographics of people with HIV in Canada would be helpful.

Response to R1.Comment2:

We have added additional details about the epidemiology of HIV in Canada and British Columbia to the introduction section, to provide better context for readers.

Added text - Introduction section (p.5):

“In Canada, there were an estimated 62,790 people with HIV (PWH) as of 2020 (0.17% of the population), of which 75.4% were men and 24.6% were women. In British Columbia (BC), the prevalence of HIV was 0.19% (n=9,637 PWH), of who approximately 82.4% were men, and 17.6% were women [12,13] . As of 2020, approximately half of PWH (50.3%) in Canada were gay, bisexual, and other men who have sex with men (gbMSM), approximately one third (32.8%) were heterosexual, and 13.3% were people who inject/injected drugs (PWID) [12]. In BC (2020), among PWH, 53.8% were gbMSM, 25.0% were heterosexual, and 17.4% were PWID. The HIV incidence in 2020 was 4.0 per 100,000 population in Canada (n=1,520 newly-identified PWH) and 2.1 per 100,000 population in BC (n=108) [12].”

R1.Comment3: 

The algorithms evaluated only included HIV encounters and hospitalizations, and did not take into account laboratory test results (e.g., viral load frequency) or ART prescriptions, although presumably these data were available. Please explain why these were not utilized in the algorithms investigated, and/or include additional algorithms that include these factors (e.g., see Ridgway et al. Comparison of algorithms for identifying people with HIV from electronic medical records in a large, multi-site database, JAMIA Open, Volume 5, Issue 2, July 2022).

Response to R1.Comment3:

Thank you for your comment, we have added explanation to these points into the discussion section. Detectable viral loads for HIV were available via the BC-CfE records in HIV registry we aimed to supplement with algorithm-identified PWH: hence we constructed healthcare use based algorithms to supplement/complement the count of PWH identified in the registry (via detectable viral loads, ART, or a positive HIV test reported to the BCCDC). HIV treatment medication was not available/included in the medication dispensing system (PharmaNet) available to researchers in BC, for our study period. We have added a note accordingly into the discussion section.

Added text, discussion section (p.24):

“Antiretrovirals, as well as detectable viral loads, are part of the HIV registry (an indicator of HIV positivity) and since we used the algorithms to supplement such indications, we focused on HIV-related healthcare use. Also, in BC, antiretrovirals for HIV were unavailable in the standard provincial medication dispensation database (PharmaNet) for our study period (data up to March 2020).”

R1.Comment4: 

The authors evaluated sensitivity and specificity of the chosen algorithms by certain demographic features (e.g., sex) but not by race. I would suggest examining the performance characteristics of the algorithms broken down by race as well, given that Black individuals are disproportionately impacted by HIV and it would be imperative to ensure that algorithms perform equally well for all racial groups.

Response to R1.Comment4:

We agree that stratification by some ethno-cultural factor would be valuable, particularly to ensure cross-group measurement equivalence; however, no well-populated indication currently exists in our datasets (this is a piece of information we hope to better populate via additional data collection moving forward, at our centre – and hence be better placed to examine its role in future work). We have added a note to this point in the discussion section.

Added text, discussion section (p.24):

“No stratifications were performed on factors related to ethno-cultural background; such information was unavailable in our datasets.”

R1.Comment5: 

The algorithms selected by the authors include the requirement for 5 HIV encounters within 12 months. This is very frequent. While PWH may have had more frequent encounters with HIV providers earlier in the study period, more recently it is not uncommon for PWH with well-controlled HIV to only see providers once or twice a year. Are the algorithms more accurate earlier in the study period? Please comment on how changes in HIV care/practice over time may impact the performance characteristics of the algorithms over time.

Response to R1.Comment5:

Thank you for this comment, we have added text to the discussion section to elaborate about the impact of different time periods on the algorithm’s performance in identifying PWH. We note that while the number of healthcare practitioner encounters (5) is relatively high, the sensitivity of over 82% suggests that a majority of PWH in our validation sub-sample did meet these criteria. We also note that we searched all available healthcare records for people’s time in BC – so even if a person has well-controlled/-managed HIV (as is the case, thankfully, for a large number of PWH in BC), they would still have been identified as PWH via our algorithm earlier in their care trajectory – perhaps in the year or so around their confirmed diagnosis via the higher level of HIV-related care associated with that event.

Added text, discussion section (p.24):

“Further, although we did not compare sensitivity/specificity by era, in the Supporting Information File it is shown that the proportion of algorithm-contributed PWH was smaller among PWH present in 2020 (2.1%; 333/8,774) versus among PWH ever present in the study period (6.2%; 985/15,957). Thus, in recent years, a larger proportion of people are identified as being PWH via the HIV registry – likely in part due to enhanced testing rates in BC, and better connections to HIV care, over time.”

Reviewer #2: 

R2.Comment1:

Specific concerns: The sample that was used for validation of the population without the disease process (without HIV), is derived from the population that has the disease process at a later date. 

Although the authors make reference to this weakness in their discussion, it should be further emphasized that the sample without HIV may not be representative of that population in the general population, and hence the performance of the algorithm may vary from what the authors present (even in that specific setting).

Response to R2.Comment1:

We have expanded on the limitations of the sample used to estimate specificity.

Expanded point in discussion section (p.23):

“Another challenge was the validation sub-sample composed of persons who all eventually tested positive for HIV in the province of BC (i.e., negative HIV test records were only available for those who had a positive HIV test on record). Therefore, the validation sub-sample without the disease (without HIV; non-cases), is sourced from the population acquired the disease at a later date. Hence, the sample without HIV may not be representative of those without HIV in the general population, and hence the algorithm’s performance (particularly, specificity) may be less generalizable than would be the case if a representative sample of the general population was used for the source of non-cases. Access to all negative HIV tests, including for individuals who never went on to test positive for HIV, would have provided a better representation of healthcare use in the period before a negative HIV test result (in our validation sub-sample).” 

R2.Comment2:

Another concern is that this sample seems to be a high utilizer of health care (mean 9 visits), which may not be representative of other areas and may alter in important ways the performance of the algorithm.

Response to R2.Comment2:

We agree that the level of healthcare utilization is relatively high, but note that the level of healthcare utilization and other characteristics are generally comparable between those included in the validation sub-sample (i.e., PWH with HIV test records available) vs those not included (the remainder of the STOP cohort), as shown in the Supporting Information File: 

Median number of all-cause outpatient practitioner encounters annually was 9.7 in the validation sub-sample vs 9.9 for the remainder of the STOP cohort. 

The STOP cohort captures the vast majority of PWH in BC who have ever accessed treatment for HIV, or have had documented indication of positivity (a detectable pVL and/or a positive HIV test result). Hence, STOP generally as well as the validation sub-sample derived from the STOP cohort is likely a reasonable representation of known PWH in BC for the observation period. The similarity/representativeness of the validation sub-sample with respect to the overall STOP cohort has been described in the manuscript, and is shown via a table in the Supporting Information File. 

Nevertheless, any healthcare-based identification algorithm will generally show better sensitivity for persons (frequently) accessing healthcare than for those with little recorded healthcare use – though specificity is also a concern if a high number of healthcare use occurs. We have added a note to the discussion section with that caveat.

Revised text, discussion (p.23)

“Study limitations exist when using case-finding algorithms based on healthcare records. Sensitivity will be better for persons with frequent documented healthcare use than for those with little/no documented healthcare use. Thus, PWH in BC without any HIV-related healthcare use recorded in BC healthcare databases will be missing from the algorithm-contributed portion of estimates of PWH. Conversely, however, high-level healthcare users may be more likely to receive false positive classifications by virtue of registering more encounters – thus attenuating specificity. We note, however, that this shortcoming is partly mitigated by the good capture of PWH via HIV registry – permitting us to prioritize specificity

R2.Comment3:

Minor issues:

Typos: Page 10, line 226; 

 page 13, line 287. 

The first row for 12 months does not align for the 24 and 36 months. 

Although it is clear why and this is explained in the Tables in the appendix, it is not explained here and it can make the reader not realize it and interpret the data in a wrong way.

Figure 3 and others: The first row for 12 months (P1H1) does not align with the first row for the 24 and 36 months (P1H2). Although it is clear why and this is explained in the Tables in the appendix, it is not explained here and it can make the reader not realize it and interpret the data in a wrong way (not everyone reviews the appendix information).

Response to R2.Comment3:

We have corrected the noted typos, and have made the table note clearer on the issue of the lack of a time window for the H1, P1 values, or the P1H1 algorithm – including it with every table/figure where algorithms are displayed.

Revised text (various locations; below relevant tables/figures)

“The H1 and P1 events were unbounded by search window since they contained single events; hence, the algorithm P1H1 referred to 1 HIV-related healthcare practitioner, or 1 HIV-related hospitalization occurring at any time.”

---

## [Decision Letter · Decision Letter 1]

15 Aug 2023

PONE-D-23-08773R1Identification of people with rare diseases in administrative healthcare records: a case study of HIV in British Columbia, CanadaPLOS ONE

Dear Dr. Emerson,

Thank you for submitting your manuscript to PLOS ONE. After careful consideration, we feel that it has merit but does not fully meet PLOS ONE’s publication criteria as it currently stands. Therefore, we invite you to submit a revised version of the manuscript that addresses the points raised during the review process. Kindly respond to Reviewer #2's comment. Please submit your revised manuscript by Sep 29 2023 11:59PM. If you will need more time than this to complete your revisions, please reply to this message or contact the journal office at plosone@plos.org. Please include the following items when submitting your revised manuscript:A rebuttal letter that responds to each point raised by the academic editor and reviewer(s). You should upload this letter as a separate file labeled 'Response to Reviewers'.A marked-up copy of your manuscript that highlights changes made to the original version. You should upload this as a separate file labeled 'Revised Manuscript with Track Changes'.An unmarked version of your revised paper without tracked changes. You should upload this as a separate file labeled 'Manuscript'.If applicable, we recommend that you deposit your laboratory protocols in protocols.io to enhance the reproducibility of your results. Protocols.io assigns your protocol its own identifier (DOI) so that it can be cited independently in the future. For instructions see: https://journals.plos.org/plosone/s/submission-guidelines#loc-laboratory-protocols. Additionally, PLOS ONE offers an option for publishing peer-reviewed Lab Protocol articles, which describe protocols hosted on protocols.io. Read more information on sharing protocols at https://plos.org/protocols?utm_medium=editorial-email&utm_source=authorletters&utm_campaign=protocols.

We look forward to receiving your revised manuscript.

Kind regards,

Edward Nicol, PhD

Academic Editor

PLOS ONE

Journal Requirements:

Reviewers' comments:

Reviewer's Responses to Questions

**Comments to the Author**

1. If the authors have adequately addressed your comments raised in a previous round of review and you feel that this manuscript is now acceptable for publication, you may indicate that here to bypass the “Comments to the Author” section, enter your conflict of interest statement in the “Confidential to Editor” section, and submit your "Accept" recommendation.

Reviewer #1: All comments have been addressed

Reviewer #2: All comments have been addressed

2. Is the manuscript technically sound, and do the data support the conclusions?

Reviewer #1: Yes

Reviewer #2: (No Response)

3. Has the statistical analysis been performed appropriately and rigorously? 

Reviewer #1: Yes

Reviewer #2: (No Response)

4. Have the authors made all data underlying the findings in their manuscript fully available?

Reviewer #1: Yes

Reviewer #2: (No Response)

5. Is the manuscript presented in an intelligible fashion and written in standard English?

Reviewer #1: Yes

Reviewer #2: (No Response)

6. Review Comments to the Author

Reviewer #1: (No Response)

Reviewer #2: I have a minor comment, in reference to one made by the other reviewer. I consider HIV to be a low prevalence disease (in the 0.2% rate in BC), but not a rare disease (by FDA definition, for the USA, a disease present in less than 200,000 individuals in the USA (about 0.05% of the current population). Somewhere in the text, the authors may want to comment in the Canadian definition of rare disease.

7. PLOS authors have the option to publish the peer review history of their article (what does this mean?). If published, this will include your full peer review and any attached files.

Reviewer #1: No

Reviewer #2: No

---

## [Author Response · Author response to Decision Letter 1]

15 Aug 2023

Dear Editorial team, Aug 15, 2023

Thank for your sharing the comment by reviewer. Initially our wording choice of ‘rare disease’ in the context of HIV was with low prevalence / low frequency in mind – not attempting to meet a particular threshold for ‘rare’. After consideration, however, we have opted to use the term ‘low prevalence’ throughout – in replacement of ‘rare disease’ to avoid potential confusion. 

Sincerely,

Scott Emerson, on behalf of the study authors

---

## [Editor Report · Decision Letter 2]

16 Aug 2023

Identification of people with low prevalence diseases in administrative healthcare records: a case study of HIV in British Columbia, Canada

PONE-D-23-08773R2

Dear Dr. Emerson,

We’re pleased to inform you that your manuscript has been judged scientifically suitable for publication and will be formally accepted for publication once it meets all outstanding technical requirements.

Kind regards,

Edward Nicol, PhD

Academic Editor

PLOS ONE
---

## [Editor Report · Acceptance letter]

23 Aug 2023

PONE-D-23-08773R2 

Identification of people with low prevalence diseases in administrative healthcare records: a case study of HIV in British Columbia, Canada 

Dear Dr. Emerson:

I'm pleased to inform you that your manuscript has been deemed suitable for publication in PLOS ONE. Congratulations! Your manuscript is now with our production department. 

Kind regards, 

on behalf of

Dr. Edward Nicol 

Academic Editor

PLOS ONE